# The Antagonism of Neuropeptide Y Type I Receptor (Y1R) Reserves the Viability of Bone Marrow Stromal Cells in the Milieu of Osteonecrosis of Femoral Head (ONFH)

**DOI:** 10.3390/biomedicines10112942

**Published:** 2022-11-15

**Authors:** Jih-Yang Ko, Feng-Sheng Wang, Sung-Hsiung Chen, Re-Wen Wu, Chieh-Cheng Hsu, Shu-Jui Kuo

**Affiliations:** 1Department of Orthopedic Surgery, College of Medicine, Chang Gung University, Kaohsiung Chang Gung Memorial Hospital, Kaohsiung 833401, Taiwan; 2Department of Medical Research, College of Medicine, Chang Gung University, Kaohsiung Chang Gung Memorial Hospital, Kaohsiung 833401, Taiwan; 3School of Medicine, China Medical University, Taichung 404328, Taiwan; 4Department of Orthopedic Surgery, China Medical University Hospital, Taichung 404327, Taiwan

**Keywords:** osteonecrosis of femoral head (ONFH), neuropeptide Y, BIBO3304, stromal derived factor-1 (SDF-1)

## Abstract

Neuropeptide Y (NPY)-Y1 receptor (Y1R) signaling is known to negatively affect bone anabolism. Our study aimed at investigating the impact of NPY-Y1R signaling in the pathogenesis of glucocorticoid-related osteonecrosis of the femoral head (ONFH). Femoral heads were retrieved from 20 patients with and without ONFH, respectively. The bone marrow stromal cells (BMSCs) from ONFH femoral heads were treated with Y1R agonists and antagonists for subsequent analysis. We showed that the local NPY expression level was lower in ONFH heads. The Y1R agonists and antagonists disturb and facilitate the survival of BMSCs. The transcription of stromal derived factor-1 (SDF-1) was enhanced by Y1R antagonists. Our study showed that the local NPY expression level was lower in ONFH heads. Y1R antagonists facilitate the survival of BMSCs and stimulate the transcription of SDF-1 by BMSCs. These findings shed light on the role of NPY-Y1R signaling in the pathogenesis of ONFH.

## 1. Introduction

Osteonecrosis of the femoral head (ONFH) is a major musculoskeletal disorder manifested by hip pain and disability and is one of the leading causes of total hip arthroplasties (THAs) [1,2]. ONFH mainly affects the young and middle-aged population and harbors an unignorable disability rate, jeopardizing the quality of life of the victims. Intense ischemic necrosis of the bone tissue, marrow edema, and decreased bone marrow stromal cells (BMSCs) are the hallmarks of ONFH [3]. The periarticular pathologic changes mentioned above could precipitate cartilage degeneration in the femoral head, ultimately leading to joint collapse [4]. In 1977, Ficat et al. proposed a classification system for ONFH, in which four stages were classified according to clinical symptoms, radiograph, magnetic resonance imaging, and isotope bone scan [5]. Ficat stage III and IV ONFH warrants THAs.

ONFH can be categorized into traumatic and nontraumatic. The main etiological factors for traumatic ONFH include femoral head and neck fracture, acetabular fracture, hip dislocation, and intraarticular hematoma compromising femoral head circulation [5]. As for non-traumatic ONFH, the pathogenesis is not completely understood, and exposure to glucocorticoid is a well-known risk factor [6,7]. There is no fidelitous way to prevent glucocorticoid-related ONFH patients from undertaking total hip surgeries at present. Although THAs harbor high patient satisfaction, hematogenous periprosthetic joint infection, periprosthetic fracture, and loosening are grave complications that cannot be eradicated at present. These unmet medical needs underscore the importance of deepening the investigation of the pathogenesis of glucocorticoid-related ONFH.

The neuropeptide-Y(NPY) is a 36-amino acid neuropeptide that has been initially recognized to mediate neuroendocrine signaling between the central and peripheral nervous systems [8]. In the central nervous system (CNS), NPY is synthesized mainly in the hypothalamus. The repertoire of central NPY functions included increasing food intake, storing energy as fat, reducing anxiety and stress, reducing pain perception, affecting the circadian rhythm, reducing alcohol intake, lowering blood pressure, and controlling epileptic seizures [9]. In the peripheral system, NPY is synthesized mostly by the neurons of the sympathetic nervous system and behaves as a strong vasoconstrictor, and contributes to the growth of fat tissues [10].

NPY exerts its function via G-protein coupled receptor subtypes named Y1R, Y2R, Y4R, Y5R, and Y6R. Central Y2R and peripheral Y1R have been proposed to regulate bone remodeling [11]. Germline NPY knockout facilitates the bone mass accumulation of mice owing to increased osteoblast activity and bone formation [12]. Oral administration of Y1R antagonist BIBO3304 for 8 weeks dose-dependently increased bone mass in mice [13]. Osteoblast-specific Y1R abrogation enhanced bone mass accumulation in mice, similar to the results of Y1R germ-line deletion, confirming the peripheral effects of Y1R on bone formation through direct action on osteoblasts [14]. These findings validated the expression of Y1R in osteoblasts and suggested that Y1R might play a negative role in bone metabolism.

Based upon the evidence that the NPY-Y1R signaling negatively affects bone anabolism, we hypothesize that the NPY pathway may be associated with the pathogenesis of glucocorticoid-related ONFH. Due to the importance of BMSC in the pathogenesis of ONFH as well as the known negative impact of NPY-Y1R signaling on bone anabolism, our study tried to investigate the impact of NPY on the ONFH head-harvested BMSCs.

## 2. Materials and Methods

Our study was approved by the Institutional Review Board of Kaohsiung Chang Gung Memorial Hospital (IRB 99-1820B) before recruiting patients and was conducted from 1 September 2010 to 31 August 2013. Every participant submitted eligible written informed consent for enrollment.

After excluding the patients with metabolic bone diseases other than postmenopausal bone loss (including hyper- or hypoparathyroidism, Paget’s disease, renal osteodystrophy) and known neoplasm history, twenty patients with end-stage (Ficat stage III and IV) glucocorticoid-related ONFH were recruited for the study. All of them had a recognized history of glucocorticoid use and underwent total hip replacement surgeries. During surgery, the hip joint was entered via the posterior approach. Osteotomy of the femoral neck was executed around 1.5 cm above the calcar. The femoral head was thus extracted, and the decomposed necrotic bone around the center of the femoral head was harvested for study. The prosthesis was implanted according to the established surgical procedures. The bone specimens around the center of the femoral head from 20 patients undergoing bipolar hemiarthroplasty for displaced femoral neck fractures without a history of alcoholism and exposure to glucocorticoid were harvested as control samples.

Histologic features of harvested femoral heads were examined using a Zeiss microscope and captured by the Zeiss Image Analysis unit (Zeiss, Oberkochen, Germany). The antibody against NPY (ThermoScientific, Rockford, IL, USA) was employed for immunohistochemistry (IHC) staining [15]. The Image pro plus 6.0 software was applied to calculate the percentage of NPY positively stained cells in each histologic slice. The serum concentrations of NPY was quantified by ELISA assay (ThermoScientific, Rockford, IL, USA).

The BMSCs from ONFH patients were harvested. Briefly, BMSCs were isolated by Ficoll-Paque^®^ (d = 1.007 g/mL, Pharmacia Biotech AB, Uppsala, Sweden) density gradient centrifugation at 500× *g* for 30 min [16,17]. Mononuclear cells were collected and seeded in Dulbecco’s Modified Eagle Medium with 10% fetal bovine serum (Life Technologies, Gaithersburg, MD, USA) in a 5% CO_2_, 37 °C incubator for 2 h. The non-adherent cells were collected and re-seeded in the 37 °C incubator for 24 h. After the non-adherent cells had been removed, the remaining adherent mesenchymal cells were harvested by trypsinization. The harvested BMSCs (1 × 10^6^ cells) were treated with the graded concentrations of Y1R agonist (Leu 31, Pro 34) (Tocris Bioscience, Bristol, UK) and Y1R antagonist BIBO3304 (Tocris Bioscience, Bristol, UK) for 24 h for subsequent experiments.

The TUNEL (terminal deoxynucleotidyl transferase deoxyuridine triphosphate (dUTP) nick end labeling)-DAPI (4’,6-diamidino-2-phenylindole) staining (Roche Diagnostics, Mannheim, Germany) was employed to illustrate cellular apoptosis after treatments. The TUNEL staining detects DNA breaks after cellular apoptosis. The TUNEL reaction mixture was prepared by mixing the enzyme solution and the label solution by a 1:9 ratio, and 50 μL TUNEL reaction mixture was added to each sample at 37 °C for one hour. After being washed with PBS solution, the TUNEL-stained samples were observed under microscopy. The morphology of nuclei was visualized by DAPI staining.

The MTT (3-(4,5-dimethylthiazol-2-yl)-2,5-diphenyl-2H-tetrazolium bromide) assay (Sigma-Aldrich, St. Louis, MO, USA) was utilized to quantify cellular viability after treatments. The MTT reagent can pass through the cell membrane as well as the mitochondrial inner membrane of viable cells. Reduction of MTT by mitochondrial reductase within viable cells resulted in the disruption of the core tetrazole ring and the formation of a violet-blue water-insoluble molecule called formazan [18]. After treating 1 × 10^6^ BMSCs with (Leu31, Pro34) and BIBO3304 under graded concentrations for 24 h, the MTT stock solution was added to each culture sample to reach the final concentration of 0.5 mg/mL for 16 h under 37 °C. The optic absorbance of formazan was measured at the wavelength of 570 nm.

The Annexin V-fluorescein isothiocyanate (FITC) apoptosis assay (BioVision, Milpitas, CA, USA) detected cellular apoptosis by discerning phosphatidylserine on the everted inner cell membrane. Propidium iodide (PI), a fluorescent DNA binding dye, freely penetrates cell membranes of dead or dying cells but is excluded from viable cells. These properties favor the use of PI for the evaluation of apoptotic cell death. The cultured BMSCs treated with Y1R agonists or antagonists were washed with cold PBS and diluted in 1X Annexin binding buffer to reach the volume of 100 μL per assay with a cell density of around 1 × 10 ^6^ cells/mL. Five μL of FITC Annexin V and 1 μL propidium iodide (100 μg/mL) working solution were added to each 100 μL of cell suspension. The cells were then incubated and stained at 25 °C for 15 min without exposure to light, and then the fluorescence emission was detected by flow cytometry.

The reverse transcription-polymerase chain reaction (RT-PCR) assay was applied to compare the transcription levels of stromal cell-derived factor-1 (SDF-1) after BMSCs were treated with Y1R agonists and antagonists. Tissues were ground under liquid nitrogen free from RNAase, and the total RNA was extracted by an RNA purification kit (RiboPure, Thermo Fisher Scientific). One microgram of extracted RNA was reverse transcribed into complementary DNA (Step One Real-Time PCR System; Thermo Fisher Scientific) following the manufacturer’s instructions. The threshold cycle (Ct) was determined as the cycle number when the fluorescence signal became detectable. The 18S ribosomal RNA (18S rRNA) was chosen as the reference gene. The ΔCt and ΔΔCt were defined by the equation
ΔCt = Ct (target gene mRNA, such as SDF-1) − Ct (18S rRNA) (1)
ΔΔCt = ΔCt(study group) − ΔCt (control group) (2)

The mRNA expression levels of target genes by the study group were expressed as 2^-ΔΔCt^ [19].

All of the values are given as the mean ± standard deviation. Between-group differences were assessed for significance using the ANOVA test, and Tukey HSD (Honestly Significant Difference) was employed for post-hoc analysis. The statistical difference was considered to be significant if the *p*-value was <0.05.

## 3. Results

There were 20 glucocorticoid-related ONFH patients (11 male, 9 female, age: 57.20 ±17.59 years) and 20 non-ONFH patients (8 male, 12 female, age: 65.80 ± 13.38 years). The composition of sex (*p* = 0.902) and age (*p* = 0.090) was comparable between the two groups.

We collected the femoral heads with ONFH (*n* = 20) from the total hip replacement surgeries and the heads without ONFH (*n* = 20) from the bipolar hemiarthroplasty surgeries for femoral neck fractures for comparative IHC analysis. Compared with the femoral heads without ONFH, the femoral heads with ONFH exhibit decreased local NPY expression (Figure 1). The serum ELISA assay also showed decreased serum NPY levels among non-ONFH patients than ONFH patients (Figure 2).

The primary BMSCs harvested from ONFH femoral heads were treated with Y1R agonist (Leu 31, Pro 34) and Y1R antagonist BIBO3304 of graded concentrations. The status of cellular apoptosis was evaluated by TUNEL-DAPI staining. TUNEL-DAPI staining (Figure 3) showed that the Y1R agonist (Leu 31, Pro 34) lead to the apoptosis of BMSCs.

The viability of ONFH-harvested BMSCs was also assessed by MTT assays and quantified by the optic density (OD). We showed that the Y1R agonist (Leu 31, Pro 34) and Y1R antagonist BIBO3304 could dose-dependently suppress and facilitate the survival of harvested BMSCs, respectively (Figure 4).

The findings of the MTT assay were subsequently validated by the Annexin V-FITC apoptosis assay. We showed that high-dose Y1R agonist (Leu 31, Pro 34) and Y1R antagonist BIBO3304 could predispose and protect the BMSCs from apoptosis, respectively (Figure 5).

Stromal cell-derived factor-1 (SDF-1) is a chemokine that could facilitate the survival and osteogenic differentiation of BMSCs [20]. We employed an RT-PCR assay to evaluate the transcription level of SDF-1 under serial concentrations of Y1R agonists and antagonists. We showed that the transcription level of SDF-1 was not affected by (Leu 31, Pro 34) but was enhanced by BIBO3304 (Figure 6).

## 4. Discussion

In our study, we showed that compared with the non-ONFH femoral heads, the ONFH femoral heads displayed decreased local NPY expression in IHC staining. Y1R agonist and antagonist showed the trend of disturbing and facilitating the survival of BMSCs, respectively. Y1R antagonist could dose-dependently enhance the transcription of SDF-1 by BMSCs. These results have not been reported before and shed light on the role of NPY-Y1R signaling in the pathogenesis of ONFH.

NPY-Y1R signaling is involved in many physiological activities, such as macrophage migration, mitogenic activity, and pulpal development [21]. The extra-CNS distribution of Y1R encompasses mainly the pancreas, intestine, and bone. Due to the substantial distribution of Y1R in bone, NPY-Y1R signaling is involved in the pathogenesis of various human musculoskeletal disorders, including osteoporosis, fracture, inflammation, and osteoarthritis [22]. In bone, Y1R is expressed by BMSCs, osteoblasts, osteocytes, monocyte/macrophage, and osteoclast, inferring the potential role of NPY on bone remodeling [22]. Y1R germline knockout has been shown to demonstrate increased osteoblast activity and mineral apposition rate, together with increased osteoclasts and increased surface area [14,23]. In Baldock et al.’s study, NPY knockout (NPY −/−) mice were presented with significantly increased bone mass, increased osteoblast activity, and enhanced expression of *Runx2* and *Osterix* than the wild-type mice. However, NPY −/− mice with hypothalamic-reserved NPY expression (AAV-NPY+) were presented with a significant reduction in bone mass compared with NPY −/− mice, but the bone loss in AAV NPY+ mice did not completely correct the high bone mass phenotype of NPY −/− mice. These data recognize NPY as a major player of bone remodeling, increasing bone mass in obesity with low hypothalamic NPY expression and decreasing bone formation under ‘starving’ with high hypothalamic NPY expression [12]. All the findings published by various authors suggested the negative impacts of NPY-Y1R signaling on bone mass maintenance.

Following exposure to insults predisposing the patients to ONFH (e.g., steroid use, alcohol abuse, thromboembolic events…), BMSCs are required for the regeneration and repair of the necrotic lesions. BMSCs possess multi-lineage potential in differentiation, and CD44 is involved in orchestrating the differentiation, proliferation, migration, and survival of BMSCs [24]. The decrease in the number of BMSCs is the hallmark change in the ONFH bone marrow [3]. In view of the decreased BMSCs in the ONFH femoral head, the application of BMSCs over the osteonecrotic lesion sites seems to be a reasonable revenue. Clinical trials of autologous BMSCs therapy for ONFH have shown promising results in early-stage cases [10,25]. However, the compromised regenerative potentials of BMSCs harvested from ONFH patients remain a problem. Recent studies suggest that endothelial dysfunction, compromised differentiation potentials of BMSCs, or aberrant angiogenesis are likely to be involved in patients with ONFH [24]. As a result, avoiding BMSCs apoptosis and facilitating the survival of BMSCs seems to be another direction worthy of investigation.

Following the thinking process of facilitating the survival of BMSCs, our study focused on the role of NPY-Y1R signaling in the pathogenesis of ONFH by elucidating the impact of Y1R agonist and antagonist on ONHF head-harvested BMSCs. We showed that the Y1R agonist and antagonist could disturb and facilitate the survival of harvested BMSCs in the MTT assay, respectively. The findings of the MTT assay could be further supported by the TUNEL-DAPI staining and Annexin V- FITC apoptosis assay. Previous publications have demonstrated the impact of Y1R signaling on primary BMSCs in bone remodeling. One recent study from Zhang’s group found that osteocytes promoted adipogenesis and inhibited osteogenesis of BMSCs by secreting NPY. Deletion of NPY expression in osteocytes generates a high bone mass phenotype and attenuates aging and ovariectomy (OVX)-induced bone-fat imbalance in mice. Osteocyte NPY production is under the control of the autonomic nervous system (ANS), and osteocyte NPY deletion blocks the ANS-induced regulation of BMSC fate and bone-fat balance. γ-Oryzanol, a clinically used ANS regulator, significantly increases bone formation and reverses aging- and OVX-induced osteocyte NPY overproduction and marrow adiposity in control mice, but not in mice lacking osteocyte NPY expression. Zhang’s study suggests a new mode of ANS control of bone metabolism via the ANS-induced expression of osteocyte NPY [26]. To explore the osteogenic differentiation capacity and the migration and vascular endothelial growth factor (VEGF) expression capabilities of BMSCs affected by NPY, Liu et al. investigated the potential relationships among NPY, osteogenic differentiation, angiogenesis and canonical Wnt signaling in BMSCs. In Liu’s study, NPY significantly promoted osteogenic differentiation of BMSCs in a concentration-dependent manner and up-regulated the expression levels of β-catenin and p-GSK-3β proteins and β-catenin mRNA. Moreover, NPY promoted the translocation of β-catenin into the nucleus. The effects of NPY were inhibited by Y1R antagonist or Wnt pathway antagonist. Additionally, NPY enhanced BMSCs migration and VEGF expression. These results suggested that NPY may stimulate osteogenic differentiation via activating canonical Wnt signaling and enhance the angiogenic capacity of BMSCs [27]. Lee et al. also showed that NPY-Y1R signaling is directly involved in the differentiation of BMSCs and the activity of mature osteoblasts. The transcription of *Runx2*, *Osterix*, and *PPAR-γ* was increased in the long bones of Y1(−/−) mice compared with wild-type mice. In vitro, BMSCs isolated from Y1(−/−) mice formed a greater number of mineralized nodules under osteogenic conditions and a greater number of adipocytes under adipogenic conditions than controls. In addition, both the number and the size of fibroblast colony-forming units formed in vitro by purified osteoprogenitor cells were increased without Y1Rs. Furthermore, the ability of the immature mesenchymal stromal cell population and a more committed osteoprogenitor cell population to differentiate into osteoblasts and adipocytes in vitro was enhanced without Y1R signaling. Finally, Y1R deletion enhanced the mineral-producing ability of mature osteoblasts, as shown by increased in vitro mineralization by BMSCs isolated from osteoblast-specific Y1(−/−) mice. Lee’s data demonstrate that the NPY-Y1R signaling cascade directly inhibits the differentiation of BMSCs as well as the activity of mature osteoblasts, constituting a likely mechanism for the high-bone-mass phenotype evident in Y1(−/−) mice [28]. Our study focused on a specific subset of BMSCs harvested from ONFH femoral heads and found that Y1R agonist and antagonist could disturb and maintain the survival of this subset of BMSCs, respectively. Our findings shed light on the impact of NPY-Y1R signaling on the viability of ONFH head-harvested BMSCs and on the pathogenesis of ONFH.

We also showed that Y1R antagonism could enhance the transcription of SDF-1 by the BMSCs. SDF-1 is a chemokine molecule that binds to its transmembrane receptor CXC chemokine receptor-4 (CXCR4). Naobumi et al.’s team published several publications discussing the impact of SDF-1 on osteogenesis. SDF-1 was co-required with bone morphogenetic protein 2 (BMP2) for differentiating mesenchymal C2C12 cells into osteoblastic cells [29]. Blocking of the SDF-1/CXCR4 signal axis or adding SDF-1 protein to BMSCs significantly affected BMP2-induced alkaline phosphatase activity and osteocalcin synthesis, markers of preosteoblasts and mature osteoblasts, respectively. Moreover, disrupting the SDF-1 signaling impaired bone nodule mineralization during terminal differentiation of BMSCs. SDF-1 blockade inhibited the BMP2-induced early expression of *Runx2* and *Osterix*. The authors thus concluded that perturbing the SDF-1 signaling affected the osteogenic differentiation of BMSCs in response to BMP2 stimulation [30]. Based upon these findings, our observations for the enhanced transcription of SDF-1 by BMSCs under the treatment of BIBO3304 shed light on the potential of enhancing osteogenic differentiation of BMSCs via Y1R antagonism.

Although NPY-Y1R signaling seems to be harmful to the survival of ONFH head-harvested BMSCs, the local expression of NPY in ONFH head is paradoxically decreased. We hypothesize that the decreased NPY expression might be the compensatory protective mechanism to mitigate the survival stress of BMSCs. However, the protective decline of local NPY expression is not sufficient to rescue end-stage ONFH head. The antagonism of NPY-Y1R signaling, not suppressing NPY local expression only, might hold promise for the treatment of ONFH.

There are several limitations to our study. Our study investigated the impact of Y1R antagonism on the bone marrow stromal cells, including a heterogenous group of cells, due to the difficulties of harvesting enough viable cells from the necrotic regions. The dedicated investigations of the impact of Y1R antagonism on a specific cell type, such as the bone marrow STEM cells, were difficult due to the scarcity of viable cells. The lack of in vivo validation for the protective effects of Y1R antagonism in established ONFH animal models is another major limitation of our study. Moreover, the implications of the paradoxically decreased systemic and local expression of NPY among ONFH patients could not be fully delineated in the present study.

## 5. Conclusion and Future Directions

Our study showed that the local NPY expression as well as systemic NPY serum levels were lower among ONFH patients. The Y1R antagonist facilitates the survival of BMSCs and stimulates the transcription of SDF-1 by BMSCs. Our study demonstrated the therapeutic potential of the employment of Y1R antagonism for the treatment of ONFH and shed light on the role of NPY-Y1R signaling in the pathogenesis of glucocorticoid-related ONFH. The protective effects of Y1R antagonism against bone cell apoptosis should be validated by in vivo ONFH animal models or even clinical studies in the future. The effects of Y1R antagonism on the osteogenic differentiation of bone marrow mesenchymal cells via the enhancement of SDF-1 expression warrants future investigation.

## Figures and Tables

**Figure 1 biomedicines-10-02942-f001:**
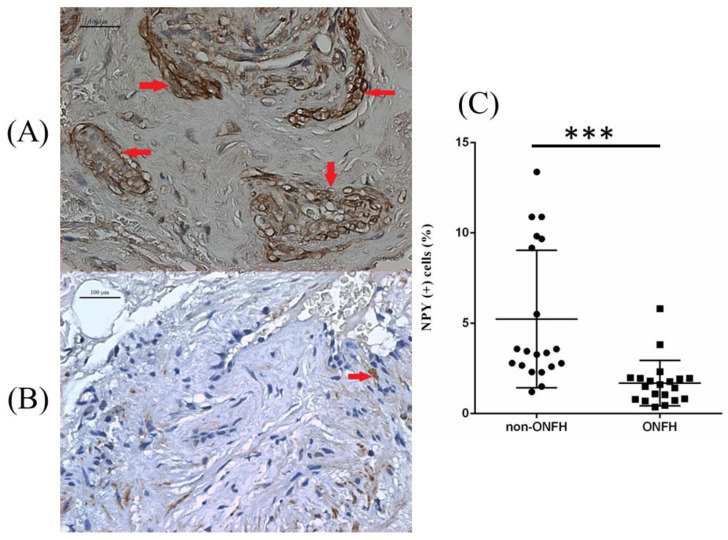
The representative illustration of NPY IHC staining of the femoral head obtained from non-ONFH (**A**) and ONFH (**B**) patients, and the percentage of NPY positively stained cells between non-ONFH and ONFH groups (**C**) (*** *p* < 0.001). The red arrows indicate positively stained cells. The black dots and the black squares indicate the information concerning the percentage of NPY (+) cells for respective patients.

**Figure 2 biomedicines-10-02942-f002:**
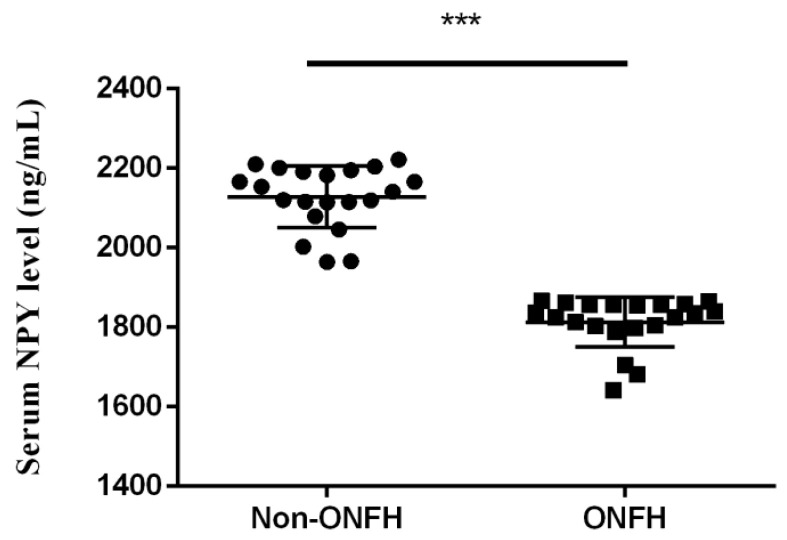
The serum NPY levels among non-ONFH and ONFH patients (*** *p* < 0.001).

**Figure 3 biomedicines-10-02942-f003:**
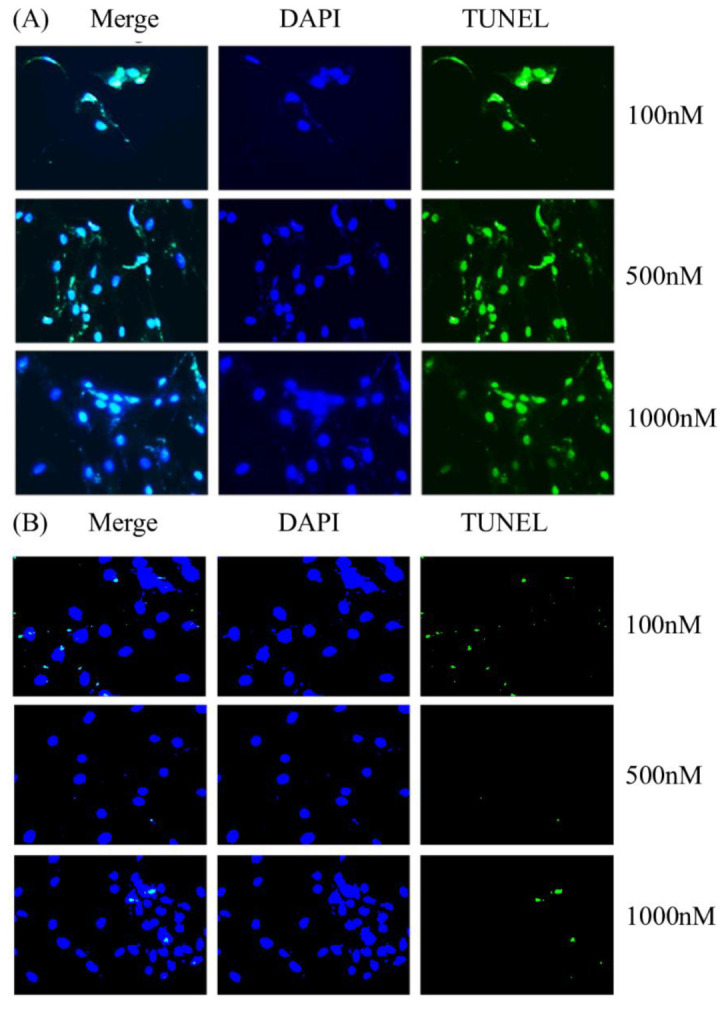
The representative illustration of DAPI-TUNEL staining of the BMSCs harvested from ONFH heads treated with Y1R agonist (Leu 31, Pro 34) (**A**) and antagonist BIBO3304 (**B**) of graded concentrations.

**Figure 4 biomedicines-10-02942-f004:**
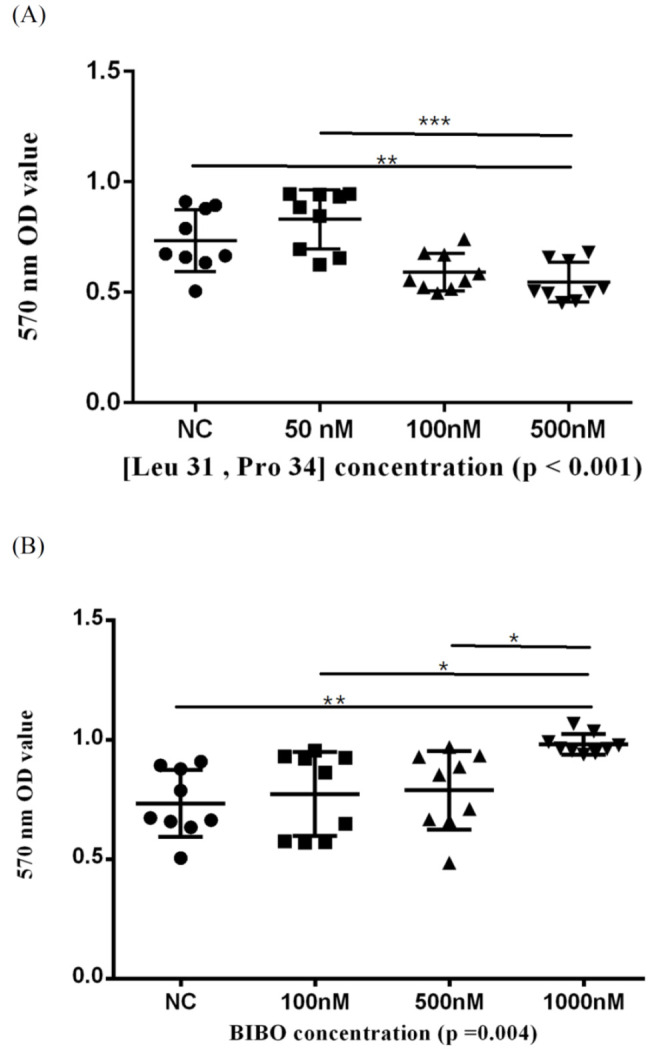
The 570 nm optic density (OD) values of BMSCs treated in serial concentrations of Y1R agonist (Leu 31, Pro 34) (**A**) and antagonist BIBO3304 (**B**) in MTT assay. Respective *p*-values for post-hoc analysis: (**A**) NC-500 nM: 0.008; 50–500 nM: <0.001; (**B**) NC-1000 nM: 0.004; 100–1000 nM: 0.018; 500–1000 nM: 0.032 (* *p* < 0.05, ** *p* < 0.01, and *** *p* < 0.001). The black triangles/dots/squares indicate the information concerning the respective 570nm OD values for respective assays.

**Figure 5 biomedicines-10-02942-f005:**
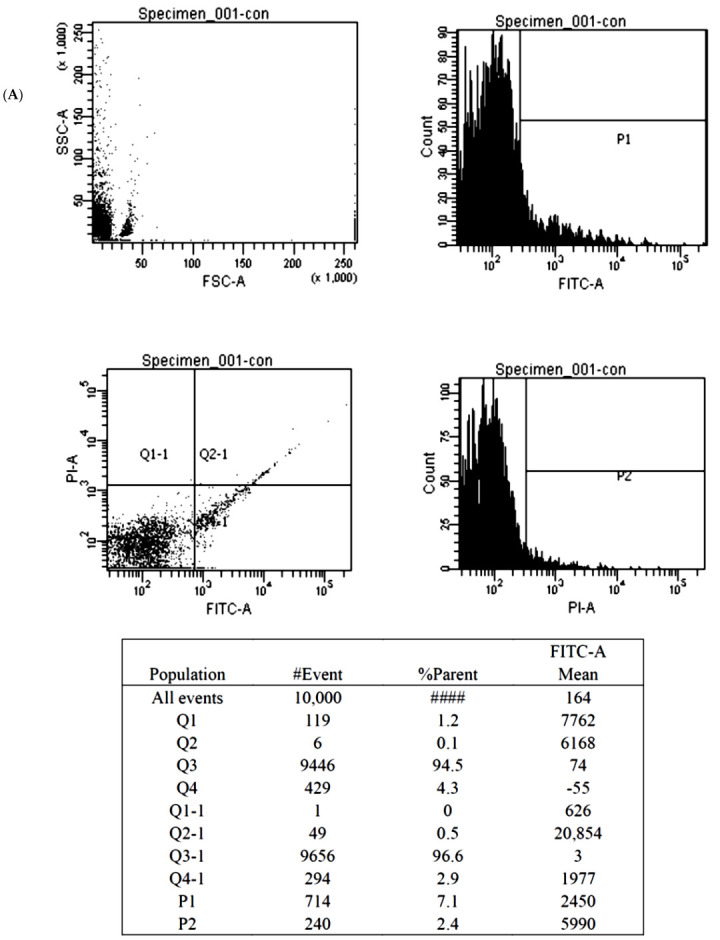
The illustration of the Annexin V-FITC apoptosis assay for the BMSCs treated by control (**A**), 1000 nM Y1R agonist (Leu 31, Pro 34) (**B**) and 1000 nM Y1R antagonist BIBO3304 (**C**). The P1 value quantified cellular necrosis, and the P2 value quantified cellular apoptosis.

**Figure 6 biomedicines-10-02942-f006:**
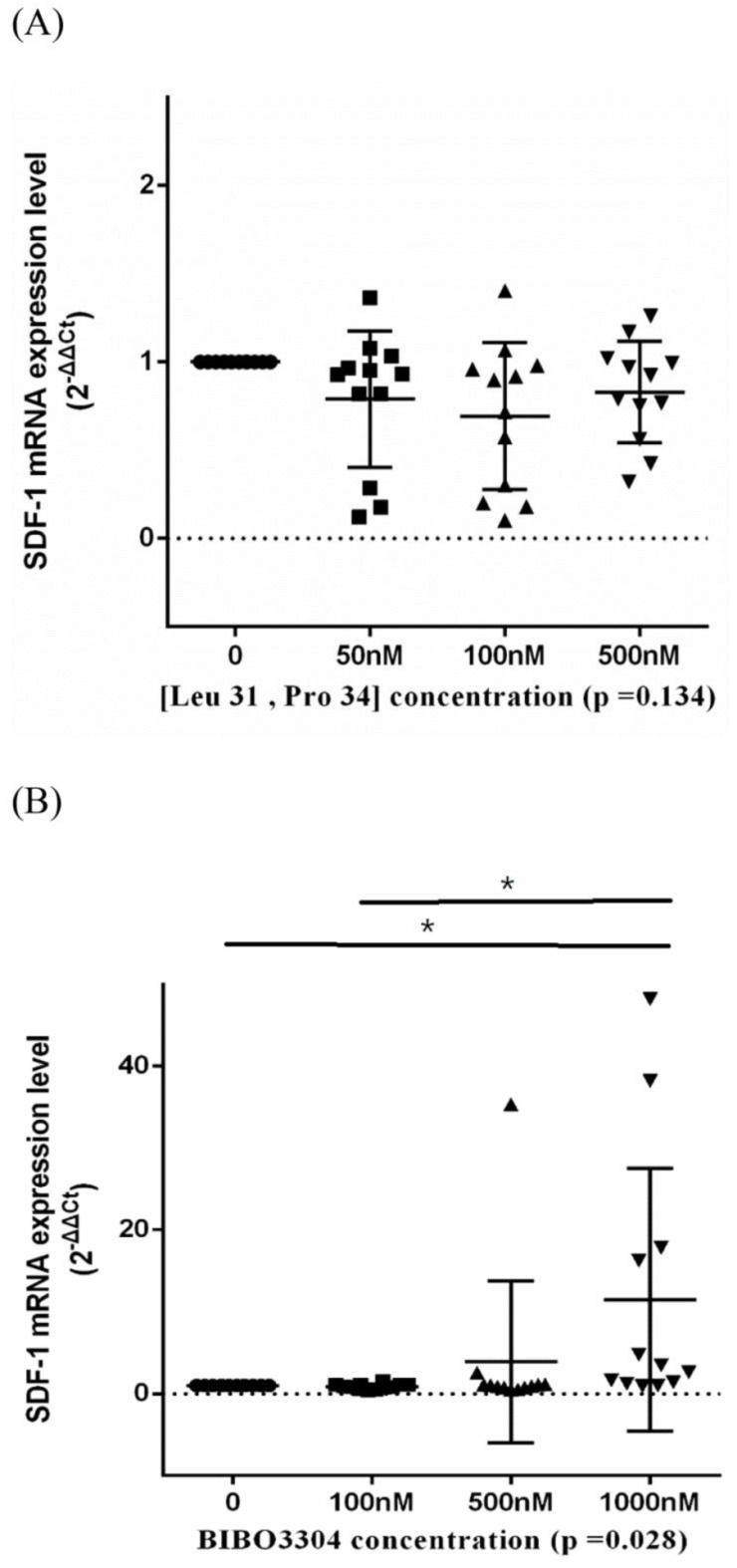
The transcription levels of stromal cell-derived factor-1 (SDF-1) of BMSCs treated in a serial concentration of Y1R agonist (Leu 31, Pro 34) (**A**) and antagonist BIBO3304 (**B**). Respective *p*-values for post-hoc analysis: (**B**) 0–1000 nM: 0.045; 100–1000 nM: 0.024 (* *p* < 0.05). The black triangles/dots/squares indicate the information concerning the respective SDF-1 mRNA expression levels for respective assays.

## Data Availability

Not applicable.

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
