# Peer review of "The Antagonism of Neuropeptide Y Type I Receptor (Y1R) Reserves the Viability of Bone Marrow Stromal Cells in the Milieu of Osteonecrosis of Femoral Head (ONFH)"

_biomedicines, 2022, doi:10.3390/biomedicines10112942_

Round 1

Reviewer 1 Report

In this investigation Ko et al. explained the Neuropeptide Y (NPY)-Y1 receptor (Y1R) antagonists facilitate the survival of bone marrow stem cells (BMSCs) and stimulate the transcription of SDF-1 by BMSCs. Also, they showed, showed that the local NPY expression level 21 was lower in osteonecrosis of the femoral head (ONFH). However, Authors need to address the following concerns before acceptance.

1)    The TUNEL-DAPI staining of the primary BMSCs in figure 2 and 3 are not clear. Can be replaced.

2)    Protein profiling studies of BMSCs pertaining to necrosis is missing. Inclusion of these results will strengthen the claim.

3)    In vivo studies of the described results would gather more readers attention.  

4)    The font size of text in some of the areas all through the manuscript was not uniform and can be rectified.

5)    The manuscript can be further revised for grammatical and typological errors.

Author Response

Thank you whole-heartedly for reviewing our manuscript. These invaluable suggestions helped us a lot in refining our work. We have revised our work following your instructions, and the changes will be detailed below:

1) The old not clear figures have been replaced by clearer figures in the new merged figure 3 (the old figure 2 and figure 3 have been merged following another reviewer’s comment).

2) Due to the scarcity of viable primary BMSCs in the necrotic lesions, the residual tissues from the recruited 20 ONFH heads do now allow for new analysis (e.g., Western blotting). We apology for this. Alternatively, we offered the findings of the “protein” quantification ELISA assay for serum NPY levels in the new figure 2.

3) The lack of in vivo findings has also been questioned by another reviewer. We apologize for this. Actually, our team try hard to establish reliable ONFH animal model but has not been reproducibly successful by now. As a result, we mention it in the limitation: “ The lack of in vivo validation for the protective effects of Y1R antagonism in established ONFH animal model is another major limitation of our study.” (line 338~340)

4) Thanks for the reminding us the issue of font size. We have checked the appropriate font size according to the template offered from the Biomedicines submission website.

5) We are not sure how many rounds of revision ahead, and we pledge for the opportunity for final language editing after the last round of revision.

Thanks again whole-heartedly for reviewing our manuscript.

Reviewer 2 Report

Overall suggestion: Accept with major changes

Major revisions:

1.     References 8-14 are really old-  has there been no advancement in the fields since? Your hypothesis is based on evidence that is not up-to-date as per the references provided by you.

2.     In general, I notice that a lot of your references (especially the first 10-15 references) are well over a decade old. While your current references can stay, I would highly encourage to add recent references from the last 5-10 years. This will further support your evidence with respect to recent advances made in the field. I have found two papers that you must add in your references but please do add more evidence as references from the last 5-10 years. (https://www.ncbi.nlm.nih.gov/pmc/articles/PMC6197539/, PMID: 30357068 and https://pubmed.ncbi.nlm.nih.gov/34067727/\ ,PMID: 34067727

3.     Lines 84-88 and lines 95-97, could you please outline – how did you extracted the bits of bone to isolate the cells? As in did you break them down into smaller fragments and then treat them with something to remove the cells? You mention the use of Ficoll which is usually done for samples in their liquid state. If you harvested the necrotic bone for your investigation – how did you get to the stage of using ficoll? How did you go from using solid bone to using Ficoll? I would have broken down the necrotic bone, then treated with collagenase and then isolated BMSCs. There is missing details in your method and if you followed a previously established/published protocol – please add their reference to make it easier for your readers/ reviewers.

4.     How did you confirm that the cells extracted were BM-MSCs? Did you phenotype them using flow cytometry? Take images of their morphology when in culture to confirm their identity? Perform the classical CFU-F assay to indicate that they are indeed bone marrow MSCs? There are many cell populations within any bone and unless we confirm the identity of the cells using at least one (of the above mentioned methods or another) technique – data may be questioned. Please provide evidence that the cells you isolated are indeed BMSCs.

Minor revisions:

5.     For figure 1 – a. Please add the scale for the images as it appears that images a and B are being presented at very different scales. Please also indicate the staining with an arrow on both the images as it is currently unclear how you have arrived at the data presented in figure c.

6.     Please combine Figures 2 and 3 in one figure to better compare the results on the effect of agonist and antagonist. Please d the same for figures 4 and 5 as well as for figures 8 and 9.

7.     For all figures – in most publications, the p values are usually denoted by * for p<0.05, **for p<0.01, *** for p<0.001 and **** for p<0.0001. Could you please add the * marks as and when indicated in your figures? You also need to write the exact p values on the figure but include them in the figure legend instead.

8.     Please add a section of ‘Limitations’ to reflect on the drawbacks of your current study and how they could be further improved. Please also re-write the conclusion section as ‘conclusion and future directions’ to emphasise on the potential application and usefulness of the study. This should include your vision for the work you have done and how you would like to take this work forward.

Author Response

Thank you whole-heartedly for reviewing our manuscript. Your penetrating but friendly comments taught us a lot. We have revised our work following your instructions, and the changes will be detailed below:

  1. We apologize for offering the old references. Some references have been exchanged by pertinent up-to-date references. However, the references from Baldock, Sousa, and Lee were kept because their works are indeed interesting , inspiring, and quite pertinent.
  2. Thank you for offering us the two excellent references. We have added the two references (reference 6 and reference 7) following your instructions respectively.
  3. We apologize for not citing the reference. Actually, the decomposed necrotic tissues are quite liquidlike and could be assessed by Ficoll. We have employed Ficoll for processing nearly liquid decomposed necrotic tissues in our previous work, and the published protocol was employed in our study and has been cited as reference 16 and 17.
  4. We apologize for the big mistake. The cells harvested following reference 17’s protocol were named as “bone marrow STROMAL cells” instead of “bone marrow STEM cells” in the original text. We indeed unsure all our harvested cells harbor proliferative and differentiative potentials. We thus change the term “STEM” cell for describing our harvested cells to “STROMAL” cells following reference 17. The following paragraph has been added to address the problem you raised : “Our study investigated the impact of Y1R antagonism on the bone marrow stromal cells, including a heterogenous group of cells, due to the difficulties of harvesting enough viable cells from the necrotic regions. The dedicated investigations of the impact of Y1R antagonism on a specific cell type, such as the bone marrow STEM cells, were difficult due to the scarcity of viable cells.” (line 335~339).
  5. The scale bar has been added in figure 1A and 1B, and the positively stained regions were marked by red arrows following your instructions. The following paragraph was added to describe how to arrive at the data presented in figure c: “ The Image pro plus 6.0 software was applied to calculate the percentage of NPY posi-tively stained cells in each histologic slice.” (line 95~96).
  6. Old figures 2 & 3 have been merged as new figure 3, old figure 4 & 5 have been merged as new figure 4, old figure 6 & 7 have been merged as new figure 5, and old figures 8 and 9 have been merged as new figure 6 following your instruction.
  7. All the asterisks in the figures have been added following your instruction, and the p values have been supplemented in figure legend.
  8. We have added limitation section: There are several limitations to our study. Our study investigated the impact of Y1R antagonism on the bone marrow stromal cells, including a heterogenous group of cells, due to the difficulties of harvesting enough viable cells from the necrotic re-gions. The dedicated investigations of the impact of Y1R antagonism on a specific cell type, such as the bone marrow STEM cells, were difficult due to the scarcity of viable cells. The lack of in vivo validation for the protective effects of Y1R antagonism in established ONFH animal model is another major limitation of our study. Besides, the implications of the paradoxically decreased systemic and local expression of NPY among ONFH patients could not be fully delineated in the present study. (line 335~343) and “ Conclusion and future directions” section : “Our study showed that the local NPY expression as well as systemic NPY serum level were lower among ONFH patients. Y1R antagonist facilitates the survival of BMSCs and stimulates the transcription of SDF-1 by BMSCs. Our study demonstrated the therapeutic potential of the employment of Y1R antagonism for the treatment of ONFH and shed light on the role of NPY-Y1R signaling in the pathogenesis of gluco-corticoid-related ONFH. The protective effects of Y1R antagonism against bone cell apoptosis should be validated by in vivo ONFH animal model or even clinical studies in the future. The effects of Y1R antagonism on the osteogenic differentiation of bone marrow mesenchymal cells via the enhancement of SDF-1 expression warrants future investigation.” (line 345 ~ 354) following your instructions. Please check whether our amendments are appropriate. Thanks again whole-heartedly for reviewing our manuscript.

Round 2

Reviewer 2 Report

Even though you have changed the term to stromal cells - it would be useful to include an image of the cells indicating their morphology for readers. however, I will leave that up to you to decide.

All of the other points have been addressed.